# Bulk Segregant Analysis Sequencing and RNA-Seq Analyses Reveal Candidate Genes Associated with Sepal Color Phenotype of Eggplant (*Solanum melongena* L.)

**DOI:** 10.3390/plants13101385

**Published:** 2024-05-16

**Authors:** Benqi Wang, Xia Chen, Shuping Huang, Jie Tan, Hongyuan Zhang, Junliang Wang, Rong Chen, Min Zhang

**Affiliations:** Wuhan Vegetable Research Institute, Wuhan Academy of Agricultural Sciences, Wuhan 430065, China; wangbenqi@wuhanagri.com (B.W.); chenxia@wuhanagri.com (X.C.); huangshuping@wuhanagri.com (S.H.); tanjie@wuhanagri.com (J.T.); zhanghongyuan@wuhanagri.com (H.Z.); wangjunliang@wuhanagri.com (J.W.); chenrong@wuhanagri.com (R.C.)

**Keywords:** eggplant, sepal color, BSA sequence, transcriptome analysis, candidate gene

## Abstract

Eggplant is a highly significant vegetable crop and extensively cultivated worldwide. Sepal color is considered one of the major commercial traits of eggplant. Eggplant sepals develop from petals, and sepals have the ability to change color by accumulating anthocyanins, but whether the eggplants in sepal and their biosynthetic pathways are the same as those in petals is not known. To date, little is known about the underlying mechanisms of sepal color formation. In this study, we performed bulked segregant analysis and transcriptome sequencing using eggplant sepals and obtained 1,452,898 SNPs and 182,543 InDel markers, respectively, as well as 123.65 Gb of clean data using transcriptome sequencing. Through marker screening, the genes regulating eggplant sepals were localized to an interval of 2.6 cM on chromosome 10 by bulked segregant analysis sequencing and transcriptome sequencing and co-analysis, combined with screening of molecular markers by capillary electrophoresis. Eight possible candidate genes were then screened to further interpret the regulatory incentives for the eggplant sepal color.

## 1. Introduction

Eggplant (*Solanum melongena* L.) holds significant agricultural value as a vital vegetable crop [1], and China is the largest eggplant producer in the world with 37,459,233.66 tons produced in 2023 (FAOSTAT; http://faostat3.fao.org, accessed on 1 December 2023). Among the various phenotypic marker traits in eggplants, differences in sepal color increase genetic diversity while enriching the market with diverse supplies to match the preferences of different consuming regions [2]. Sepals, integral components of the flower structure, play a pivotal role in safeguarding other floral organs, such as stamens, pistils, and petals, as typically, sepals envelop the exterior of the flower [3]. Understanding the genetic patterns governing sepal color in eggplant is crucial, both theoretically and practically, this knowledge is crucial in developing eggplant varieties that meet commercial quality standards and align with diverse market demands [4,5]. Enhancing fruit quality in eggplants presents a significant challenge for agronomists, with sepal color emerging as a pivotal attribute. Notably, the preference for eggplant sepal color exhibits regional variations, with green sepal eggplants gaining popularity in China [6]. Traditional breeding methods face limitations in efficiently selecting cultivars with desired sepal colors. Manual detection by researchers post-flowering is not only labor-intensive, but also time-consuming. Sepal color is an important agronomic trait of eggplant plants that directly affects the appearance characteristics of varieties. Studying the genetic laws and molecular mechanisms of sepal color can provide theoretical support and technical means for the genetic improvement of eggplant. This research can accelerate the breeding process, and enhance the yield and quality of eggplant.

To date, investigations into the inheritance and regulatory mechanisms of eggplant sepal color have been limited. Earlier studies categorized eggplant sepal into three colors: purple, greenish-purple, and green [7]. A comprehensive analysis of fruit traits across four generations (P1, P2, F1, and F2) revealed sepal color as a quantitative trait governed by two additive-dominant-epistatic major genes, along with additive-dominant-epistasis of polygenes. The genetics of flower and fruit color have been extensively explored in the Solanaceae family, including potato, sweet pepper, and tomato, where anthocyanin synthesis and accumulation significantly influence plant color formation [3,8,9]. In the specific context of eggplant, the study identified seven traits associated with eggplant content, controlled by one to six quantitative trait loci (QTL), with four loci specifically linked to sepal color in the mapping population [10]. Recent efforts involved a genome-wide association analysis to uncover the genetic basis for eggplant pigmentation and fruit color. This led to the identification of 12 loci spanning nine chromosomes [11,12]. Notably, one gene, *Sme2.5_01638_g00005.1*, was identified as encoding a candidate flower anthocyanidin synthase (FAS) [13]. These findings reveal the molecular process of eggplant coloration and provide valuable implications for future breeding strategies and genetic operations to pursue the ideal color of eggplant sepal.

Bulked segregant analysis (BSA) is an elegant technique for identifying DNA markers that are closely linked to the causal gene of a particular trait. This is achieved by analyzing bulked samples from two populations with extreme phenotypes [14,15]. The advancement of genome resequencing technology has enabled high-throughput screening of single nucleotide polymorphisms (SNPs) in complex plant genomes [16,17,18]. The application of genome resequencing to BSA, also known as QTL-seq, has demonstrated powerful capabilities in identifying QTLs in various plant species, including rice, cucumber, and watermelon. However, the implementation of QTL-seq necessitates expensive equipment and reagents [19,20]. To enhance the resolution and precise location of primary QTLs, scientists often complement their studies with transcriptome analysis of differentially labeled genes. Transcriptome sequencing (TSE), a second-generation sequencing platform, allows for the rapid and comprehensive acquisition of almost all transcripts and gene sequences from a particular cell or tissue of a species in a specific state. This technology can be utilized to study gene expression, gene function, structure, variable splicing, and the prediction of new transcripts [21,22]. Currently, researchers widely use transcriptome sequencing in basic research, clinical diagnosis, and drug discovery.

Anthocyanin synthesis primarily regulates the color of eggplant sepals. These water-soluble pigments reside in the vesicles of plant epidermal cells, imparting various hues to plant tissues and organs. Sepals predominantly exhibit green, purple, and intermediate hues, and these color variations are primarily a result of the synthesis of pigments within the eggplants [23,24] Variations in eggplant coloration, observed in the rinds and sepals, stem from discrepancies in anthocyanin content and types [1]. Over time, the metabolic synthesis pathway of anthocyanins in eggplants has been categorized into three stages. Firstly, phenylalanine is converted into coumaroyl coenzyme A through enzymatic reactions. Secondly, enzymatic reactions akin to those involving dihydroflavonol-4-reductase lead to the synthesis of dihydroflavonols and flavanones. Lastly, stable anthocyanins form through subsequent enzymatic reactions [11,25]. These anthocyanins, also known as anthocyanidin glycosides, play a pivotal role in determining flower color. The six main anthocyanins include cyanidin, delphinidin, pelargonidin, peonidin, petunidin, and malvidin [26].

Recent discoveries have revealed two general types within the anthocyanin synthesis pathway. One type involves mutations in enzymes along this pathway, leading to isolated and characterized mutants. The other type encompasses mutations in structural genes within the anthocyanin synthesis pathway. In most instances of these mutants, the regulatory genes are affected, and these regulatory genes bear transcriptional similarities to the MYC and MYB gene families [27]. Given that anthocyanins primarily reside in the pericarp and sepals of eggplants, the coloration of eggplant sepals is intricately linked to the synthesis and quantity of anthocyanins [28]. Increasing anthocyanin synthesis in the pericarp and sepals during eggplant fruit development causes the sepals to assume a purple or dark purple color. A decrease or blockage in anthocyanin synthesis results in the sepals taking on a green or light green hue [29]. Transcriptional regulation of the anthocyanin biosynthesis pathway is orchestrated by transcription factors (TF) at the molecular level [30]. The proteins R2R3-MYB, bHLH, and WD40 control multiple enzymatic steps in the biosynthetic pathway of flavonoids, which are important secondary metabolites and precursors for anthocyanin biosynthesis across various plant species [30,31,32].

In this study, we took an active approach by identifying master QTL genes responsible for regulating eggplant sepal color. This involved constructing an F2 segregating population based on eggplant sepal color phenotypes. We employed screening techniques, including BSA sequencing and capillary electrophoresis, and integrated them with transcriptomics analysis. Our primary objectives were (1) to screen potential master QTL candidate genes associated with eggplant sepal color regulation and (2) to explore the potential regulatory mechanisms of anthocyanin biosynthesis in eggplant sepals. The results offer valuable insights into the formation and biosynthesis of sepal color in eggplant, contributing to our understanding of the molecular mechanisms governing sepal color development.

## 2. Materials and Methods

### 2.1. Plant Materials and Phenotyping for Sepal Color

Two eggplant inbred lines, PP05 (purple sepal, purple fruit, and purple flower) and GP05 (green sepal, purple fruit, and purple flower) were used as parental lines to develop segregating populations for sepal color. A cross was made between PP05 (female parent, P1) and GP05 (pollen donor, P2) to create F1, which was self-pollinated to generate the F2 population, and backcrossed with PP05 to generate for BC1P1 or with GP05 for BC1P2. Sepal color of F2 plants was recorded in three experiments conducted in Wuhan with 200, 255, and 285 individuals, respectively. Sepal color of the BC1P1 and BC1P2 populations, as well as P1, P2, and F1 plants, were investigated in all experiments. Measurement of sepal color was performed according to the methods described in [33]. All the plant materials mentioned above were grown in the nursery garden in Wuhan Vegetable Institute, China. The investigation of fruit development was performed with the average daily maximum temperature of 36.2 °C and the average daily minimum temperature of 22.5 °C, respectively.

### 2.2. DNA Extraction

Genomic DNA was extracted from fresh sepals of two parental plants and F2 generation plants using the CTAB method [34]. The extracted DNA was assessed by electrophoresis on 1% agarose gel and spectrophotometry using the NanoDrop 2000 spectrophotometer (Thermo Scientific, Wilmington, DE, USA). This study prepared four DNA samples, comprising two parental samples (GP05 and PP05) and two F2 pools (BSA-green and BSA-purple). Parental samples were prepared by separately mixing DNA from five GP05 and PP05 plants. F2 pools were constructed by mixing an equal amount of DNA from 30 green and 30 purple sepal F2 plants separately. A standard input of 1.5 μg DNA was used for each sample preparation.

### 2.3. Library Preparation and Illumina Sequencing

Sequencing libraries were constructed using Truseq Nano DNA HT Sample Preparation Kit (Illumina Inc., San Diego, CA, USA) in accordance with the manufacturer’s instructions. Briefly, The DNA samples were ultrasonically cut into fragments of approximately 350 bp in size, and then the DNA fragments were end-complemented to base A, ligated with full-length junctions for Il-lumina sequencing, and further amplified by PCR. Finally, PCR products were purified (AMPure XP system, Beckman Coulter, Brea, CA, USA) and the size distribution of the libraries was analyzed with an Agilent 2100 Bioanalyzer (Agilent Technologies Inc., Santa Clara, CA, USA) and used for real-time PCR quantification. These libraries were sequenced by Illumina Hiseq 2500 platform and 150 bp paired-end reads were generated with insert size around 350 bp. The library construction, sequencing, and data processing were accomplished at Wuhan Frasergen Bioinformatics Co., Ltd. (Wuhan, China).

### 2.4. Bulk Segregant Analysis Sequencing

DNA samples from 30 eggplant individuals of each phenotype were mixed in equal amounts to obtain one green mixed pool and one purple mixed pool. Four DNA libraries were, thus, constructed and sequenced on the Illumina HiSeq X Ten platform (Illumina, CA, USA). The raw data were then filtered using fastp software v0.23.4 to remove low-quality reads and adaptor sequences. Reads were mapped to the most recent reference genome of eggplant using HISAT2 (https://solgenomics.net/ftp/genomes/Solanum_melongena_V4.1/ (accessed on 1 December 2023)). Single-nucleotide polymorphisms (SNPs) and insertion/deletion mutations (InDels) were analyzed using Genome Analysis Toolkit (GATK) software (version 4.5.0.0). The average distribution of SNPs analyzed on the 12 chromosomes of eggplant was then calculated using a sliding window analysis with a window size of 1 Mb and an increment of 100 kb. Δ (SNP index) values were calculated based on the difference in SNP indices between the two gene libraries, as described previously. Candidate regions were selected by determining 95% confidence intervals [19].

### 2.5. RNA Extraction and cDNA Library Construction

We extracted total RNA from newly grown sepals of green and purple eggplant lines using the TRIzol kit (Invitrogen, Waltham, MA, USA) and purified it using the RNA purification kit (Promega, Madison, WI, USA) according to the manufacturer’s instructions. A total of nine cDNA libraries with three biological replicates per sample were constructed using the Illumina TruSeq RNA Sample Preparation Kit (Illumina, San Diego, CA, USA) according to the manufacturer’s instructions [35]. Each cDNA library was evaluated using the Agilent 2100 Bioanaylzer (Agilent Technologies, Santa Clara, CA, USA) and the ABI StepOnePlus Real-Time PCR System (Applied Biosystems, Waltham, MA, USA). The concentration and quality of each cDNA library were assessed and tested, respectively. The constructed and tested libraries were then sequenced on the Illumina HiSeq X Ten platform (Illumina, San Diego, CA, USA).

### 2.6. RNA-Seq Data Analysis

We processed and analyzed the RNA-Seq raw data by first removing low-quality reads, reads containing adaptor sequences, and reads with high content of unknown bases (Ns). Clean reads from each sample were then mapped to the reference genome sequence of the eggplant by HISAT2. Only uniquely mapped reads were considered for gene expression analysis. Differential gene expression and transcript abundance (expressed as fragments per kilobase per million mapped reads (FPKM) values) were finally calculated using the RESM program 1.3.0 [36]. Genes with FPKM < 1 in all samples were excluded from subsequent analyses. Differentially expressed genes (DEGs) were identified using DESeq2 based on two criteria: false discovery rate (FDR) < 0.01 and |log2fold change (FC)|>1 [37].

### 2.7. Functional Annotation of DEGs

In this study, we focused on searching the Gene Ontology (GO) and Kyoto Encyclopedia of the Genome (KEGG) functional annotations of DEGs, using blast2go3 and blastx/blastp for the GO database and KEGG database, respectively [38]. Typically, studies with *p* values ≤ 0.0001 for GO terms and Q values ≤ 0.05 for KEGG pathways are considered significant.

### 2.8. Quantitative Real-Time PCR Validation

Total RNA was extracted from the same material as the transcriptome sequenced in this study for qRT-PCR validation. Eight genes related to phenylalanine metabolism and eggplant metabolism were then selected to validate the accuracy of the RNA sequencing data using the experimental method of qRT-PCR [39]. In this study, the primer pairs were designed using Primer 5.0 (Thermo Fisher, Waltham, MA, USA).

Details of the primer pairs used are given in Appendix A. qRT-PCR was performed on an ABI StepOne™ Real-time PCR System (Applied Biosystems, Waltham, MA, USA). All qRT-PCR experiments were designed to include three technical and three biological replicates. The eggplant actin 2 gene was used as an internal control.

## 3. Results

### 3.1. Eggplant Sepal Color and Population Construction

In this preliminary study, we employed the eggplant cultivars GP05 (green sepal eggplant) and PP05 (purple sepal eggplant) as parental lines to generate F2 segregating populations through crosses and self-pollination techniques in greenhouse conditions. We established a genetic population comprising 30 plants each of P1, P2, F1, and 240 plants of the F2 segregating population, which were grown and maintained under conventional conditions. Harvesting of eggplants occurred upon reaching commercial fruit maturity, and identification was based on the specifications for describing eggplant germplasm resources. Sepal colors were categorized as green, mixed green, purple-green, mixed purple, and purple. The parental lines GP05 and PP05, as well as the subgenerational hybrid pools of green sepal eggplants (BSAg gc) and purple sepal eggplants (BSAp pc), underwent high-throughput sequencing using the Hiseq2500 platform (Figure 1A,B).

### 3.2. BSA Analysis and Initial Localization of the Primary QTL

To identify candidate genes associated with the eggplant sepal color phenotype, we conducted BSA-Seq analysis using BC1F2 plants as materials. After screening, we generated a total of 123.65 Gb of clean data, wherein 29.63 Gb, 29.45 Gb, 32.88 Gb, and 31.65 Gb corresponded to the green mix pool, the purple mix pool, the green parental pool, and the purple parental pool, respectively (Figure 1C, Appendix A). The Q30 ratio and GC content of each pool exceeded 93.90% and 36.9%, respectively. Within the green and purple pools, we identified a total of 1,452,898 SNPs and 182,543 InDels (Appendix A). We proceeded to utilize SNP loci showcasing genotypic differences between the two mixing pools. COG annotation of genes in SNPs within the candidate region revealed that the following three pathways were most enriched, including post-translational modification, protein turnover, chaperones translation, ribosomal structure, and biogenesis signal transduction mechanisms (Appendix A). The depth of each base in different mixing pools was counted, and the ED value for each locus was calculated. To mitigate background noise, the raw ED values underwent multiplication, and the median + 3SD of the fitted values for all loci was established as the association threshold for analysis, resulting in a calculated value of 1.53. In accordance with this association threshold, we identified one region with a total length of 19.80 Mb (Figure 1D,E). The position of markers on the genome enabled us to fit the delta SNP-index value of markers on the same chromosome using the DISTANCE method. Subsequently, based on the association threshold, regions above the threshold were selected as those associated with the trait. Through computer simulation experiments, at a confidence level of 99, we obtained a total of two regions encompassing a length of 77.20 Mb. The regions associated with the trait were determined by the intersection of the results obtained through the two association analysis methods. In summary, the association regions of SNPs obtained through both association analysis methods were considered, and the intersecting interval was identified as the interval ranging from 62,600,000 to 82,400,000 on chromosome 10. We applied the same method to calculate InDel markers, and following the association analysis of the two markers, we localized the candidate genes within the intervals of chromosome 10, specifically, 62,600,000–67,250,000 and 68,150,000–82,400,000, constituting a total size of approximately 18.92 Mb (Figure 2A). Subsequently, we designed molecular markers within this interval and conducted the screening of differential markers using capillary electrophoresis (Appendix A). Through this screening process, we identified eight SSR markers and one CAPS marker exhibiting co-segregation in F2 segregating populations (Figure 2B and Appendix A). The physical distance of the candidate region was estimated to be about 2.62 Mb (Appendix A), encompassing a total of 409 annotated gene (Appendix A).

### 3.3. RNA-Seq Assembly, Unigene Annotation, and DEG Analysis

In order to deeply further investigate the genetic and regulatory mechanisms of eggplant sepal color phenotype, we harvested three independent biological replicates of parental materials pc and gc, as well as samples from F2 segregating populations of GP05 and PP05 strains, and took samples of newly sprouted sepals for transcriptome sequencing analysis. After first filtering and analyzing the raw data, 20.17, 20.55, 19.95, and 20.26 Gb of clean data were achieved average for gc, pc, GP05, and PP05, respectively (Appendix A). More than 78.5% of these pure reads were mapped to unique eggplant genomic positions and the uniquely mapped reads were used for further analysis. Pearson’s correlation coefficient analyses of the three replicates in the experiment showed consistency, indicating that the RNA-Seq results are highly reliable (Appendix A). To visualize the distribution of FDR values and differentially folded FC values of all genes in the two sets of samples, we plotted MA plots and volcano plots of GP05 vs. PP05 comparisons (Figure 3A). Principal component analysis (PCA) is also commonly used to assess between-group differences and within-group sample duplications, using linear algebra calculations for dimensionality reduction and principal component extraction of RNA-seq gene variables, and we performed PCA analysis of gene expression values (FPKM) for all samples (Appendix A).

In the sepals of GP05 and PP05, we detected 46,851 and 43,759 expressed genes, respectively. Using a Venn diagram, we quantified the number of differentiated genes, revealing 1114 highly overlapping genes (Figure 3B). Subsequently, we conducted two pairwise comparisons of gene transcript levels: GP05 vs. PP05, focusing on the differences in sepal color observed in the field. Figure 3C illustrates the distribution of DEGs uniquely expressed in each comparison and those expressed in two or more comparisons. Comparing GP05 vs. PP05, we observed 1356 up-regulated genes and 1603 down-regulated genes with |log^2^ fold| change > 1. Additionally, when comparing gc vs. pc, we identified 55 up-regulated genes and 94 down-regulated genes (Figure 3C).

### 3.4. Gene Ontology and KEGG Enrichment Analyses of DEGs

In order to further investigate the functional significance of the differential genes, our study involved conducting GO analysis and KEGG enrichment analysis to categorize the screened differential genes into three main areas: biological processes, cellular components, and molecular functions (Figure 4A). In the comparison of GP05 and PP05, we conducted GO analysis of DEG expression differential genes to elucidate gene expression associated with eggplant sepal color, aiming to identify gene regulatory processes linked to anthocyanin synthesis and regulation. The analysis showed that within biological processes, metabolic processes, cellular processes, and single organism processes emerged as the primary enrichment types. Regarding cellular component processes, cellular, cellular fraction, and membrane fraction stood out as the most dominant enrichment types. Lastly, among molecular function enrichment types, catalytic activity, binding activity, and transporter activity surfaced as the major enrichment types. The KEGG pathway enrichment analysis demonstrated high consistency with the GO analysis. When compared to PP05, we observed that the down-regulated genes in GP05 were primarily enriched in phytohormone signaling, the MAPK signaling pathway, and the phenylpropanol biosynthesis pathway. Conversely, the up-regulated genes showed significant enrichment in the carbon metabolism, the MAPK signaling pathway, and the phenylpropanol biosynthesis pathway. The KEGG enrichment analysis of DEGs identified in the GP05 and PP05 comparison highlighted the importance of phytohormone signaling, the MAPK signaling pathway in eukaryotes, and phenylpropanoid biosynthesis organisms as crucial and common pathways influencing anthocyanin synthesis (Figure 4B). Derived mainly from the aromatic amino acid phenylalanine and, to a lesser extent, from tyrosine in monocotyledonous plants, anthocyanins are an important class of plant secondary metabolites. Anthocyanins are widely distributed in the plant kingdom and play an important role in all aspects of plant growth and development. They are integral components of the cell wall, protect against strong light and ultraviolet radiation, and mediate plant–pollinator interactions. (Figure 4C).

### 3.5. Association Analysis between BSA-Seq and RNA-Seq Data

To rapidly identify candidate genes associated with the green sepal phenotype in eggplants, we conducted association analysis by integrating BSA-Seq and RNA-Seq results. Utilizing the transcriptome, BSA data, and capillary electrophoresis screening of molecular markers, we screened a total of 409 genes in the candidate regions related to the sepal color phenotypes. Among these, eight candidate genes potentially involved in regulating eggplant sepal color were identified. These genes predominantly belong to the categories of anthocyanin synthesis and regulated expression (Table 1). The identified genes include transcription factors with bHLH, MYB, and WD domains crucial for anthocyanin synthesis regulation, as well as chalcone-flavonone isomerase, a key enzyme in anthocyanin synthesis. Additionally, vesicular transport proteins and several RNA transcriptional modification proteins were identified as potential regulators of eggplant sepal color. To further investigate, we compared the expression of these candidate genes in the purple sepal parent (pc) and green sepal parent (gc) using qRT-PCR assay. The analysis revealed significant differences in the expression of the screened candidate genes in both parents. The comparison showed that in *SMEL4.1_10g016510.1*, *SMEL4.1_10g017230.1*, *SMEL4.1_10g019120.1*, and *SMEL4.1_10g019310.1*, the expression was significantly higher in purple sepals than in green sepals, while *SMEL4.1_10g018710.1* exhibited the opposite result, with higher expression in green sepals. From this, we deduce that several of the above genes may be closely related to eggplant sepal color (Appendix A).

## 4. Discussion

### 4.1. Eggplant Sepal Color Is Important in Breeding for Eggplant Diversity

In recent years, with the increasing demand from consumers, breeders have progressively focused on enhancing fruit appearance quality. Fruit organ color, as a pivotal phenotypic trait and a potential linkage marker for resistance or strain, has garnered significant attention, prompting active investigations into its inheritance pattern [2,40]. Sepals are important commercial phenotypic markers and among the dominant markers in eggplant breeding, where noticeable differences are apparent. Eggplant fruits are not only edible but also favored by many consumers for a wide variety of culinary applications [41]. Varieties with purple skin typically have purple sepals, while those with green and white skin exhibit green sepals. The introduction of foreign eggplant varieties in recent years has brought forth a multitude of green, black, and purple skin varieties, enhancing consumer choices. However, it has also raised the bar for breeders, demanding higher commodity quality [42]. This experiment systematically analyzed and studied the genetic characteristics of eggplant sepal color, aiming to provide a theoretical foundation for further unraveling the genetic laws governing eggplant sepal color and expediting the breeding process to enhance eggplant’s commercial appearance quality.

### 4.2. Eggplant Metabolism Plays a Key Role in the Color of Eggplant Sepals

Eggplant is an important crop that is grown and consumed in many countries [43]. Eggplant varieties with dark purple skin and green sepal color are more attractive to consumers [44]. Purple sepal eggplant varieties have higher anthocyanin concentrations compared to other darker colored fruits and vegetables. In brilliant flowering plants, color changes in the floral organs are closely associated with anthocyanins natural pigments that are widely found in plants [45]. They not only confer plants’ vivid colors to attract pollinators and seed dispersers, but also enhance plants’ resistance to stress [45]. Anthocyanin synthesis and transport are intricately regulated processes that engage a multitude of structural genes and transcription factors, along with regulatory mechanisms at various levels, including RNA processing and protein modification [46]. The enzyme chalcone synthase (CHS) initiates the pathway by catalyzing the conversion of coumaroyl coenzyme A and malonyl coenzyme A into chalcone, which significantly influences the regulation of flower color [47]. Additionally, the transport of anthocyanins is facilitated by several transporter proteins that assist in the movement of synthesized anthocyanidin glycosides from the cytoplasm to the vacuoles for accumulation [48]. Proteins that are involved in the processing of RNA and the modification of proteins also play crucial roles in the post-transcriptional regulation of anthocyanin synthesis [49]. For instance, miRNAs exert control over the expression of genes involved in anthocyanin synthesis by targeting them for degradation or by repressing their translation in *Capsicum annuum*. The analysis in this study identified the corresponding chalcone synthases (*SMEL4.1_10g016630.1*) and genes coding for RNA processing and modifying proteins (*SMEL4.1_10g019120.1, SMEL4.1_10g019460.1*) associated with anthocyanin synthesis and transport.

### 4.3. Involvement of Transcription Factors in Phycocyanin and Biosynthesis and Regulation in Plants

The anthocyanin biosynthesis of eggplants in plants is controlled by structural genes, but also by regulatory genes and other factors [50]. Transcriptional regulation affects anthocyanin synthesis by altering transcription rates and, thus, gene expression, controlling when transcription occurs and the amount of RNA produced [51]. Transcription factors play an important role in the process of transcriptional regulation. A transcription factor (TF) is a protein that binds to specific DNA sequences, either alone or in complex with other proteins, and either enhances or blocks the recruitment of specific genes to the RNA polymerase enzyme, which regulates gene expression [52]. Transcription factors are characterized by the presence of one or more DNA-binding domains (DBDs), through which they bind to DNA sequences in the vicinity of genes, thus accomplishing regulation. The anthocyanin synthesis pathway is mainly regulated by four transcription factor families, namely MYB, bHLH, WD40-repeat, and bZIP, and it has been shown in the Arabidopsis anthocyanin synthesis pathway [53,54]. AcMYBF110 plays an important role in the regulation of anthocyanin accumulation by specifically activating the promoters of several anthocyanin pathway genes in kiwifruit pulp color regulation [55]. However, co-expression of AcbHLH1, AcbHLH4, or AcbHLH5 with AcMYBF110 induced greater anthocyanin accumulation in tobacco leaves and actinomycetes than AcMYBF110 alone [56]. In this study, MYB transcription factor (*SMEL4.1_10g016510.1*), bHLH49 (*SMEL4.1_10g018220.1*), and WD (*SMEL4.1_10g019310.1*), which are significantly differentially expressed and associated with probable anthocyanin synthesis, were screened out by comparison.

## 5. Conclusions

In this study, we conducted a primary localization of genes regulating sepal color in eggplants. Firstly, we constructed a segregated population based on eggplant sepal color and observed that eggplant sepal color exhibits quantitative traits. We categorized sepal color into five classes for differentiation. The F2 segregation population comprised 30 extreme green and purple sepals. We subjected the parental material to BSA analysis and transcriptome analysis. Through molecular marker screening, we located the main effector gene of the QTL regulating sepal color in an interval of 18.92 Mb on chromosome 10, which was subsequently shortened to an interval of 2.62 Mb by molecular marker screening (10_77683697-10_80307823). Within this interval, there were 409 genes. Through comparative annotation and expression analysis, along with transcriptome data, we discovered that the regulation of eggplant sepal color is primarily attributed to the control of the anthocyanin synthesis pathway. Consequently, we identified eight genes related to anthocyanin synthesis, including transcription factors with bHLH, MYB, and WD domains crucial for anthocyanin synthesis regulation, chalcone-flavonoid isomerase as a key enzyme in anthocyanin synthesis, and vesicle transport proteins, along with several RNA transcriptional modifiers. This identification helps further elucidate the factors triggering the regulation of eggplant sepal color and provides valuable reference data for studying the function of eggplant genes.

## Figures and Tables

**Figure 1 plants-13-01385-f001:**
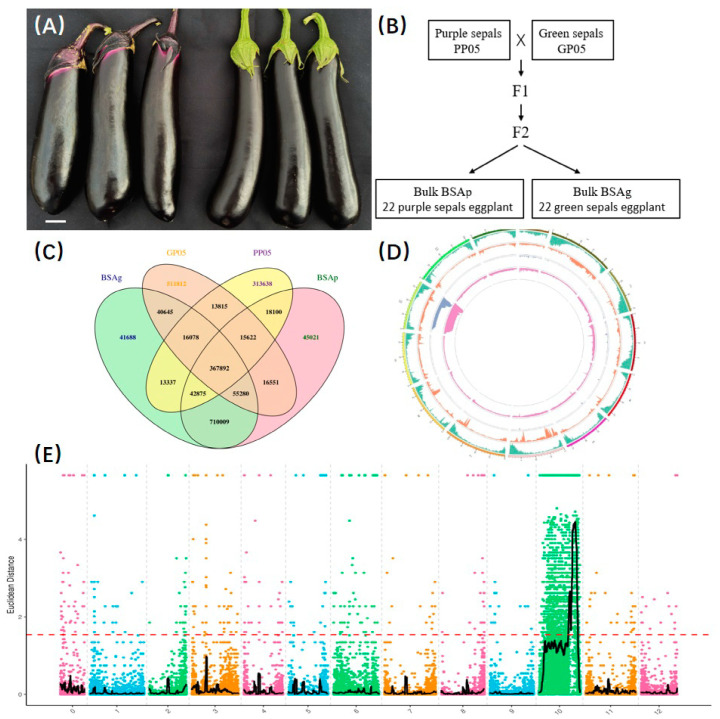
(**A**) Eggplant parents’ phenotype, (**B**) schematic diagram of population construction and bulk pool sequencing. (**C**) Venn diagram of InDel statistics between samples, (**D**) Inter-sample results visualize distribution on chromosomes (SNP), (**E**) distribution of ED association values on chromosomes.

**Figure 2 plants-13-01385-f002:**
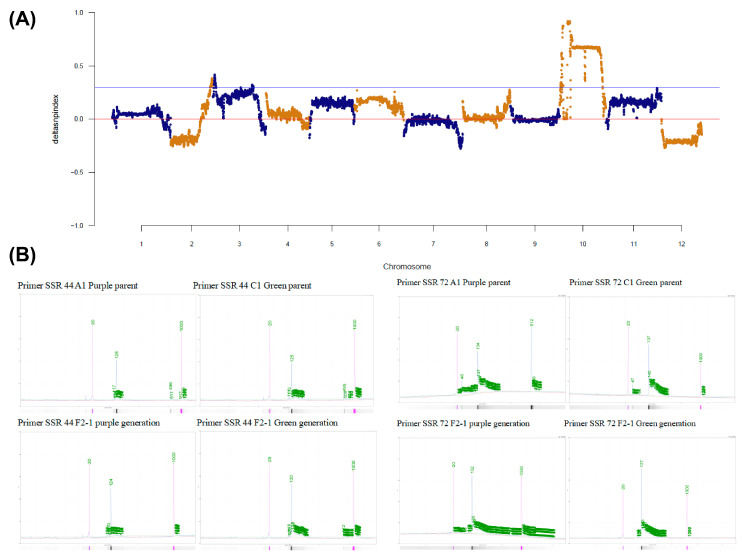
Calculation of delta SNP index and screening of molecular markers. (**A**) Delta SNP index computes the candidate intervention, (**B**) capillary electrophoresis screening for molecular marker differences between parental and extreme single plants.

**Figure 3 plants-13-01385-f003:**
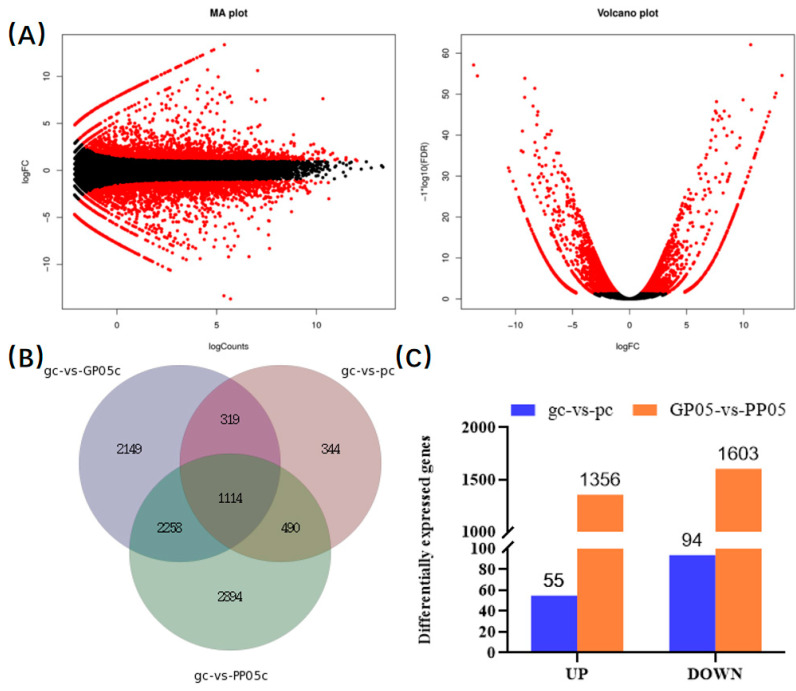
Analysis of transcriptome differences. (**A**) GP05 vs. PP05 gene expression MA plot and volcano plot, (**B**) differential gene Venn diagram schematic, (**C**) gc vs. pc and GP05 vs. PP05 significant differential expression of gene statistics.

**Figure 4 plants-13-01385-f004:**
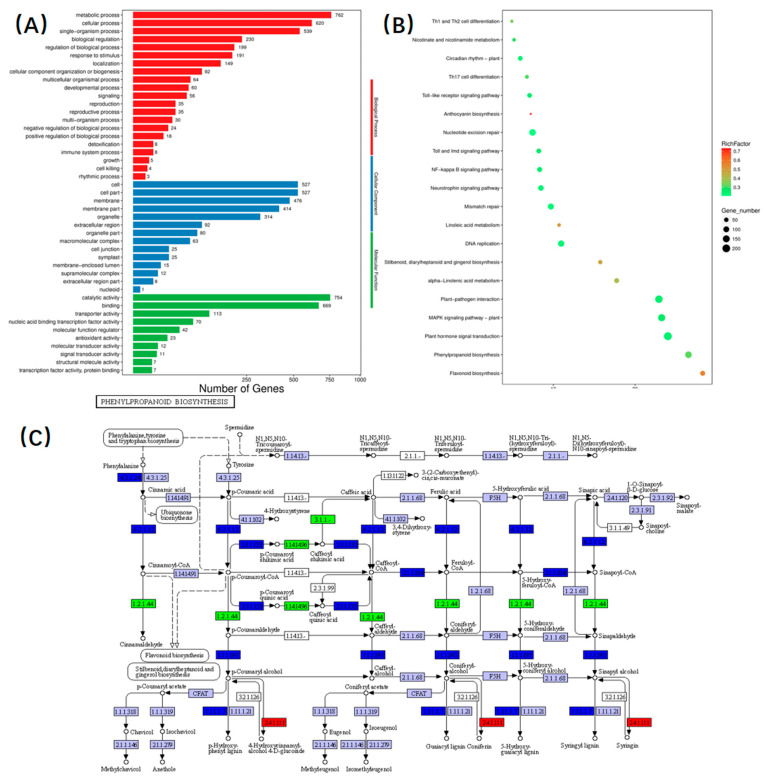
(**A**) Gene GO categorization histogram (GP05 vs. PP05), (**B**) top 20 enriched KEGG terms of DEGs were identified in various comparisons (GP05 vs. PP05), (**C**) significant enrichment of KEGG pathway metabolic pathway.

**Table 1 plants-13-01385-t001:** Candidate genes that may be related to sepal color in eggplant.

Gene Number	Gene Annotation
SMEL4.1_10g016510.1	Transcription factor bHLH49
SMEL4.1_10g016630.1	Chalcone synthase
SMEL4.1_10g017230.1	Vacuolar-sorting receptor 1
SMEL4.1_10g018220.1	MYB family transcription factor
SMEL4.1_10g018710.1	SRF-type transcription factor
SMEL4.1_10g019120.1	RNA processing and modification
SMEL4.1_10g019310.1	WD domain, transcription factor
SMEL4.1_10g019460.1	RNA processing and modification

## Data Availability

Data are contained within the article and Appendix A.

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
