# Peer review of "Bulk Segregant Analysis Sequencing and RNA-Seq Analyses Reveal Candidate Genes Associated with Sepal Color Phenotype of Eggplant (Solanum melongena L.)"

_plants, 2024, doi:10.3390/plants13101385_

Round 1
Reviewer 1 Report
Comments and Suggestions for Authors
Rigor is essential in scientific language. Considerations about the importance of the role and colour of sepal must be restricted at minimum. It is not necessary to emphasize the importance of sepal colour along the draft because oversize the focused matter. That’s can be revised from the beginning. Examples in pr 10, 27, 39 and so on… must be rewrite along the paper (except in the very concise paragraph 4.1)
Many references are not confident with the asseveration referenced. Examples in ref 4, 11, 14, and so on. Please review the convenience of the references along the draft
At the end, the authors want to indicate the group of eight candidate genes (of anthocyanin metabolism, transcription factor…) associated with the QTLs. And then, based in expression analysis, it was chosen those genes as involved in the sepal colour expression. It is used the SNP index (20) and a cosegregation in F2 to find the region where must be controlled the trait, but then the final group of genes are selected based in RNA seq and homology tests. It is surprising that cosegregation of each gene are not studied. The used method can indicate some genes that can be involved. But ones it is made the selection, it is mandatory to achieve complementation tests to check that each gene cosegregate with the trait. That will be easy with the populations constructed previously (it has suitable from F2 to BC population) and there are really efficient genomics methods to evaluate in the populations.
If the test to evaluate cosegregation doesn’t identify those genes, It will be better to rewrite the final part talking more about functions with genes that may be involved without selecting genes, but just mentioning them.
Comments on the Quality of English LanguageSome paragraph shows some dificulties to understand, mainly in the description of the gene selection methods, but it is a convenient use of english.
Author Response
Rigor is essential in scientific language. Considerations about the importance of the role and colour of sepal must be restricted at minimum. It is not necessary to emphasize the importance of sepal colour along the draft because oversize the focused matter. That’s can be revised from the beginning. Examples in pr 10, 27, 39 and so on… must be rewrite along the paper (except in the very concise paragraph 4.1) Many references are not confident with the asseveration referenced. Examples in ref 4, 11, 14, and so on. Please review the convenience of the references along the draft.
Answer: Thank you to the teacher for providing scientific and reasonable suggestions. We have carefully reorganized and rewritten the relevant expressions in the full text, including lines 10, 27, 39, etc., and we will pay more attention to related issues. We have also verified and revised the entire references more accurately to determine, which includes references 4,11,14,26,30 etc., in order to be more precise and more scientific and reasonable for the citation of the article.
At the end, the authors want to indicate the group of eight candidate genes (of anthocyanin metabolism, transcription factor…) associated with the QTLs. And then, based in expression analysis, it was chosen those genes as involved in the sepal colour expression. It is used the SNP index (20) and a cosegregation in F2 to find the region where must be controlled the trait, but then the final group of genes are selected based in RNA seq and homology tests. It is surprising that cosegregation of each gene are not studied. The used method can indicate some genes that can be involved. But ones it is made the selection, it is mandatory to achieve complementation tests to check that each gene cosegregate with the trait. That will be easy with the populations constructed previously (it has suitable from F2 to BC population) and there are really efficient genomics methods to evaluate in the populations.
Answer: Thank you very much for your questions and suggestions. we utilized BSA sequencing, transcriptome sequencing, and molecular marker-assisted selection to pinpoint the main QTL loci that regulate eggplant sepal color within a 2.62 Mb interval. Subsequently, we analyzed the transcriptome data to identify eight candidate genes within this interval that may be associated with eggplant sepal color. For the subsequent phase of our research, we plan to expand the localization population to more precisely map the primary QTL for eggplant sepal color. Following this, we will perform transgenic validation on the genes that have been identified in the regulation of sepal color. This ongoing work is dedicated to dissecting the molecular mechanisms that regulate eggplant sepal color. However, it's important to note that the candidate genes identified in this study are predictive, and we have already conducted qRT-PCR analysis on them. In our forthcoming studies, we will concentrate our efforts on these genes, employing both forward and reverse genetic verification methods. This dual approach will aid us in not only locating the specific genes but also in understanding the regulatory mechanisms that control the sepal color in eggplants.
Reviewer 2 Report
Comments and Suggestions for Authors
This study shows relevant results related to the identification of genes controlling the sepal colour in eggplant. Authors carried out a highly effective research strategy to determine candidate genes. The interest of the study lies in increasing the knowledge about molecular mechanisms governing sepal colour. Authors integrated results from bulked segregant analysis (BSA) in a segregant population for high-throughput sequencing and BSA-Seq; the combination of data permitted to identify an interesting region in chromosome 10. The design of molecular markers and qRT-PCR validation in parental lines revealed 8 interesting genes related to anthocyanin synthesis.
The manuscript is clear and well written, introduction is complete and updated. Material and methods were clearly explained. In plant material, authors indicate the process of population development and size, that are appropriate. Methodology for DNA and RNA extraction and sequencing process as well as qPCR validation was carefully described. Results was very organized and clear; figures and tables provide excellent information. Authors provide an interesting discussion. I think this is an outstanding study.
Minor corrections:
-Figure 1: In the text of figure letters D and E should be in bold
-Lane 358: the word “anthocyanins” was written in a different font size
-Reference 1, cited in the text in line 26 should reveal the importance of the crop, I think a reference related to official statistics as FAOstat web page or similar, should be more appropriate.
-Please review italics for genus and species in references
Author Response
This study shows relevant results related to the identification of genes controlling the sepal colour in eggplant. Authors carried out a highly effective research strategy to determine candidate genes. The interest of the study lies in increasing the knowledge about molecular mechanisms governing sepal colour. Authors integrated results from bulked segregant analysis (BSA) in a segregant population for high-throughput sequencing and BSA-Seq; the combination of data permitted to identify an interesting region in chromosome 10. The design of molecular markers and qRT-PCR validation in parental lines revealed 8 interesting genes related to anthocyanin synthesis.
The manuscript is clear and well written, introduction is complete and updated. Material and methods were clearly explained. In plant material, authors indicate the process of population development and size, that are appropriate. Methodology for DNA and RNA extraction and sequencing process as well as qPCR validation was carefully described. Results was very organized and clear; figures and tables provide excellent information. Authors provide an interesting discussion. I think this is an outstanding study.
Answer:Thank you for your affirmation and encouragement, we will work harder to continue the follow-up research work.
Minor corrections:
-Figure 1: In the text of figure letters D and E should be in bold.
Answer:Thank you for your suggestion, we have changed the D and E in the text to bold and we will pay more attention to the writing style of the article.
-Lane 358: the word “anthocyanins” was written in a different font size
Answer:Thank you for your suggestion, we have changed "anthocyanins" to the same font.
-Reference 1, cited in the text in line 26 should reveal the importance of the crop, I think a reference related to official statistics as FAOstat web page or similar, should be more appropriate.
Answer:Thank you for your suggestion to make our article more scientific and reasonable. After detailed inquiry, we added (FAOSTAT; http://faostat3.fao.org) statistical data of eggplant production in China to make the article more convincing.
-Please review italics for genus and species in references
Answer: Thank you for your suggestions, we have checked and revised the formatting in the references in detail, especially the italicization of species and genera.
Reviewer 3 Report
Comments and Suggestions for Authors
The presented work aims to propose genes involved in the sepal color in eggplants.
My major criticism is, that it is not clear why the color of the sepals in eggplants are so crucial for commercialization. However, the basics of the approach seem OK but some issues need either more detailed explanations or additional experiments. I will be able to point out only few:
A) In the chapter “RNA-Seq assembly, unigene annotation, and DEG analysis” no RNA-seq assembly nor an annotation process is described. It is unclear which samples were sequences for RNA-seq since they describe the sampling of pc and gc lines but later have results of gc, pc, GP05, and PP05, respectively. It is also not clear the comparison of genes according to the expression data and according to foldchanges (DEGs). Also, the strategy of the analysis comparing the different data is not explained (Figure 3). A major problem I also see is that only sepal samples were analyzed, and no control is included, such as leaves of other parts of the fruit, to be clearly able to emphasize the choice of the key genes for the sepal color.
B) In the chapter “Gene ontology and KEGG enrichment analyses of DEGs,” it is not clear which genes were used for the downstream analysis. Also, the results shown in Figure 4 are not explained in detail (Figure 4C is missing an explanation of the color code) to understand the importance of the results. In GO for example, the authors mention very generic terms such as biological processes, cellular components, and molecular functions, which are irrelevant for a proper description of the results of an analysis.
C) In the chapter “Association analysis between BSA-Seq and RNA-Seq data,” the process of the extraction of the genes of interest is not clearly described. Further, additional research on the possible genes regulated by the selected TF is missing and could give better hints about the function of those genes.
C) The Discussion and the Conclusions are kind of a repetition of the introduction with little knowledge from the results added.
Comments on the Quality of English Language
The English is very difficult to understand, there are many typos, mistakes with upper- and lower case, missing or too many spaces, problems with punctuation, and incomplete sentences.
Author Response
The presented work aims to propose genes involved in the sepal color in eggplants.
My major criticism is, that it is not clear why the color of the sepals in eggplants are so crucial for commercialization. However, the basics of the approach seem OK but some issues need either more detailed explanations or additional experiments. I will be able to point out only f
- A) In the chapter “RNA-Seq assembly, unigene annotation, and DEG analysis” no RNA-seq assembly nor an annotation process is described. It is unclear which samples were sequences for RNA-seq since they describe the sampling of pc and gc lines but later have results of gc, pc, GP05, and PP05, respectively. It is also not clear the comparison of genes according to the expression data and according to foldchanges (DEGs). Also, the strategy of the analysis comparing the different data is not explained (Figure 3). A major problem I also see is that only sepal samples were analyzed, and no control is included, such as leaves of other parts of the fruit, to be clearly able to emphasize the choice of the key genes for the sepal color. A)
Answer: Thank you for your valuable comments and suggestions for our article, which we identified and rewrote during the writing process. First of all the transcriptome sequencing was parental pc and gc, along with GC05 and GP05 of the F2 segregating population, which we corrected in the article. Since in this study we chose material with very small differences in leaf, stem, pericarp color, and fruit shape of the parents to reduce background differences, and the differences in the F2 population were mainly in sepal color, with very little other differences, we took only the sepals that had the most significant differences as cross-references in the sequencing analysis.
- B) In the chapter “Gene ontology and KEGG enrichment analyses of DEGs,” it is not clear which genes were used for the downstream analysis. Also, the results shown in Figure 4 are not explained in detail (Figure 4C is missing an explanation of the color code) to understand the importance of the results. In GO for example, the authors mention very generic terms such as biological processes, cellular components, and molecular functions, which are irrelevant for a proper description of the results of an analysis.
Answer:Thanks to your suggestions for our article, we added differential genes in Table S8 in the Supplementary file table. In addition, we also added the interpretation of the colors in Figure 4C " Red indicates proteins annotated to only up-regulated genes, green indicates proteins annotated to only down-regulated genes, and blue indicates proteins annotated to both up- and down-regulated genes ". Finally, we showed the number of genes enriched in these pathways mainly in the analysis of biological processes, cellular components and molecular functions mentioned in the GO analysis, thus resolving the metabolic pathways mainly involved in this shape difference.
- C) In the chapter “Association analysis between BSA-Seq and RNA-Seq data,” the process of the extraction of the genes of interest is not clearly described. Further, additional research on the possible genes regulated by the selected TF is missing and could give better hints about the function of those genes.
Answer: Thanks to your suggestion, we have provided a more detailed representation of the screening process for the predicted 8 candidate genes, aiming to enhance readers' understanding of the article. Additionally, we included a depiction of the role of transcription factors in anthocyanin synthesis, along with references to related studies.
- C) The Discussion and the Conclusions are kind of a repetition of the introduction with little knowledge from the results added.
Answer:Thank you for your suggestion, based on your comments, we have revised both the discussion and conclusion sections by adding more discussion and derivation to make our article more scientific and reasonable.
Comments on the Quality of English Language
The English is very difficult to understand, there are many typos, mistakes with upper- and lower case, missing or too many spaces, problems with punctuation, and incomplete sentences.
Answer:Thank you for your suggestions, we have revised the article in detail and also corrected typos, caps errors and so on. Thank you for your input to make our articles more standard and correct.
Round 2
Reviewer 1 Report
Comments and Suggestions for Authors
Authors acepted the corrections suggested. It follow other minor corrections. After addressing this, paper may be accepted.
Line 270: It couldn’t find the data described in table S2
Lines 271 to 275 : It couldn’t find the functional pathways described, in figure S1
Just one tipology: fig 2 uses delta snp index and line 281 uses ΔSNP-index
Line 318: GP05-GP05 seems to be a mistake
Line 372 : Seems more proper use colour sepal phenotype
Line 382: finish with a point
Author Response
Line 270: It couldn’t find the data described in table S2
Answer: Thank you for your question, this data is in our Table S3 data and is re-labelled in the article of line 270, thank you for asking questions to complete our article.
Lines 271 to 275: It couldn’t find the functional pathways described, in figure S1
Answer: Thank you for your suggestions for our article, Figure S1 is a CAPS-labelled agarose gel plot, which we illustrate and cite at the end of section 3.2 of the article, lines 255-257.
Just one tipology: fig 2 uses delta snp index and line 281 uses ΔSNP-index
Answer: Thank you for giving us a question to make our article more correct, we have changed it in line 281 of the article to be consistent with what is shown in the picture.
Line 318: GP05-GP05 seems to be a mistake
Answer: Thank you for giving us an error on the article, it was indeed my mistake that led to the error, it should have been GP05-PP05, we have corrected it in the article.
Line 372: Seems more proper use color sepal phenotype
Answer: Thank you for your suggestion for us to make our article more scientific and reasonable. We have made a change in line 372 and indeed it is more accurate to read "sepal colour phenotype" than "green sepal phenotype".
Line 382: finish with a point
Answer: Thanks to your suggestion, we have rewritten "From this, we deduce that several of the above genes may be closely related to eggplant sepal color" in line 382 of the article to summarise the results and make our article more reasonable and accurate.